# In situ small-angle X-ray scattering reveals strong condensation of DNA origami during silicification

Martina F. Ober[1], Anna Baptist[2], Lea Wassermann[2], Amelie Heuer-Jungemann[2] ✉ & Bert Nickel[1] ✉

Silicification of DNA origami structures increases their stability and provides chemical protection. Yet, it is unclear whether the whole DNA framework is embedded or if silica just forms an outer shell and how silicification affects the origami's internal structure. Employing in situ small-angle X-ray scattering (SAXS), we show that addition of silica precursors induces substantial condensation of the DNA origami at early reaction times by almost 10 %. Subsequently, the overall size of the silicified DNA origami increases again due to increasing silica deposition. We further identify the SAXS Porod invariant as a reliable, model-free parameter for the evaluation of the amount of silica formation at a given time. Contrast matching of the DNA double helix Lorentzian peak reveals silica growth also inside the origami. The less polar silica forming within the origami structure, replacing more than 40 % of the internal hydration water, causes a hydrophobic effect: condensation. DNA origami objects with flat surfaces show a strong tendency towards aggregation during silicification, presumably driven by the same entropic forces causing condensation. Maximally condensed origami displayed thermal stability up to 60 °C. Our studies provide insights into the silicification reaction allowing for the formulation of optimized reaction protocols.

DNA origami[1] is a versatile bottom-up nanofabrication technique to engineer nanometer-sized objects with sub-nanometer precision and complete site-specific addressability due to the programmable self-assembly of complementary DNA strands[2]. Potential applications of such DNA origami objects are manifold and include bio-sensing[3], drug delivery, as well as various biophysical[4] and biomedical applications[5–9]. A major bottleneck of utilizing DNA origami nanostructures in biomedical applications, however, is their inherent instability in common biological buffers and cellular environments as well as their susceptibility to enzymatic degradation[10–12]. Therefore, there is a need to increase the chemical, thermal and mechanical stability of DNA origami nanostructures in order to unravel their full potential and utilization in real-life applications.

One recently reported approach to achieve higher stability of DNA origami nanostructures is their encapsulation in a protective silica shell. Resulting structures are even stable in the absence of salt-containing buffers, at high temperatures, and in the presence of nucleases[7,13,14]. We demonstrated silicification of single DNA origami nanostructures and 3D DNA origami crystals[15], resulting in mechanical enforcement. This stabilization allowed us to analyze these fragile origami structures in the dry state, without suffering from structural collapse[13,16]. Silicified DNA origami structures are promising candidates for biomedical applications and they play a prominent role for the customized synthesis of inorganic dielectric 2D[17,18] and 3D nanomaterials[7,19,20].

The silicification process is a sol-gel approach based on a modified Stöber reaction[7,13,14]. The reaction is initiated through the electrostatic

[1]Faculty of Physics and CeNS, Ludwig-Maximilians-Universität München, Geschwister-Scholl-Platz 1, 80539 Munich, Germany. [2]Max Planck Institute of Biochemistry and CeNS, Ludwig-Maximilians-Universität München, Am Klopferspitz 18, 82152 Martinsried, Germany.
✉e-mail: heuer-jungemann@biochem.mpg.de; nickel@lmu.de

interactions of the quaternary ammonium head group of N-tri-methoxysilylpropyl-N,N,N-trimethylammonium chloride (TMAPS) and the anionic DNA phosphate backbone. Siloxane groups on TMAPS then provide co-condensation sites for tetraethyl orthosilicate (TEOS) and enable silica growth. The successful growth of silica on DNA origami nanostructures was thus far mainly evidenced through analysis of structures in the dry state via transmission electron microscopy (TEM), scanning electron microscopy (SEM), atomic force microscopy (AFM) and energy dispersive X-ray spectroscopy (EDX)[7,13,14]. "Shell" thick-nesses were inferred indirectly through microscopy images. However, to date it is unclear how the silicification reaction commences and whether silica grows as a "shell" around the origami, or if silica also penetrates the internal structure of the helix bundles. In view of many possible applications of silicified-DNA origami nanostructures, espe-cially as sculptured dielectrics, detailed understanding of the internal structure is essential in order to rationalize the protective nature of the silicification and its dielectric properties. Nevertheless, conventionally applied microscopy and spectroscopy techniques do not allow for such detailed investigation and analysis. Diffraction techniques such as small-angle X-ray scattering (SAXS) provide nanoscale information on DNA origami[12,15,21] and silica nanocomposites at physiological condi-tions in solution[22,23].

In this work, we employ in situ SAXS to study the silicification process. We reveal and quantify a TMAPS-induced condensation of the inner double helix spacing of 24 helix bundles (24HBs) and four-layered origami bricks (4-LBs), as well as an outer shape contraction. Silica forms both on the inside and outside of the DNA origami as revealed by X-ray contrast matching. The inner order of the origami and the overall shape are well-preserved. We demonstrate that silica penetration into the origami structure is the main cause for increased thermal stability up to 60 °C rather than an outer silica shell. Moreover, we observe that DNA origami with flat surfaces show increased ten-dency towards aggregation during silicification.

## Results

From previous reports, it is known that DNA silicification is a slow process, taking at least several hours, often up to 7 days[7,13,14]. Here we followed the silicification process via an X-ray lab source using Mo characteristic radiation[24]. Mo X-rays induce less radiation dose com-pared to Cu radiation of the same intensity[25], allowing for long in situ SAXS experiments with drastically reduced radiation damage to the sample. Furthermore, Mo radiation allows for larger absorption lengths along the beam (10 mm vs. c.a. 1.5 mm) yielding more practical geometric constrains for SAXS sample cells. As DNA origami objects exhibit a tendency to sediment during silicification, we constructed a special cell allowing for tumbling of the sample with -1 round/s around its centre to ensure well-dispersed DNA origami solutions throughout the measurement (see supporting information Supplementary Note 2 for details).

The silicification reaction was continuously analyzed by SAXS measurements. These measurements are then binned in time to achieve the best signal-to-noise ratio. We found that a binning time of 1 h was sufficiently fast to follow the silicification reaction with good X-ray statistics.

Prior to silicification, a reference measurement of the purified origami was taken. The SAXS intensity distribution for the bare 24HBs is shown in Fig. 1a. The SAXS signal I(q) exhibits three distinct intensity oscillations with dips at $q \approx 0.05$ Å$^{-1}$, $q \approx 0.09$ Å$^{-1}$, and $q \approx 0.13$ Å$^{-1}$. These dips are characteristic for the cylindrical shape of 24HBs. Modelling of the 24HB as a homogeneous cylinder[12] with radius $R_{bare} = 80.1 \pm 0.2$ Å allowed matching of the SAXS intensity in this q-range. At $q \approx 0.16$ Å$^{-1}$, the SAXS intensity shows an additional, Lorentzian-shaped peak, which is not predicted by the homogeneous cylinder model. In order to reproduce this feature, the structure model was extended by the

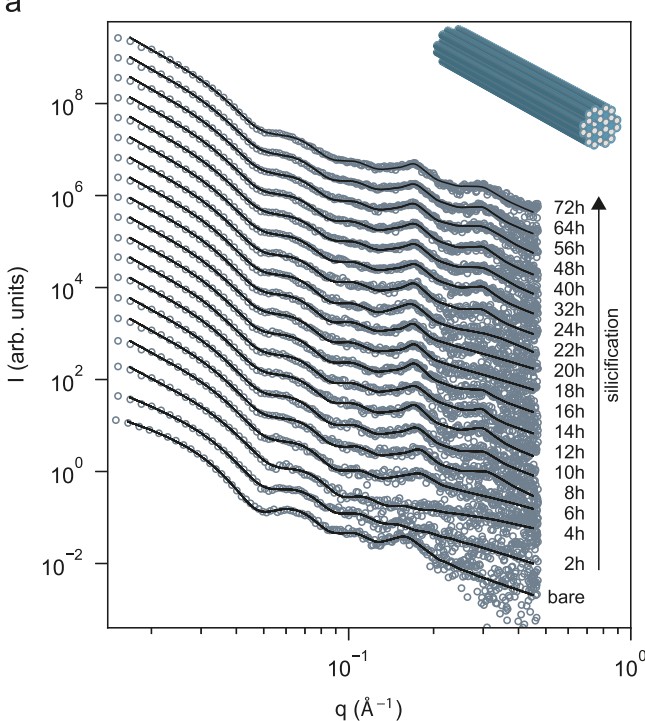

a

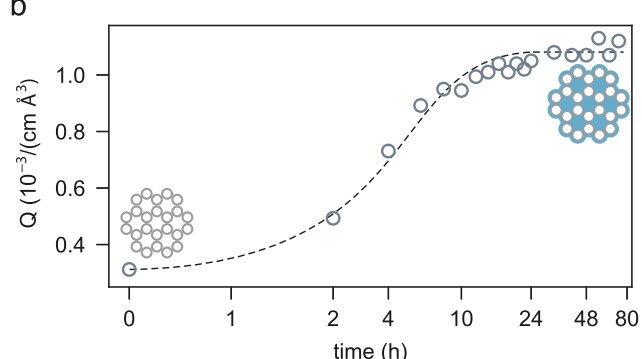

b

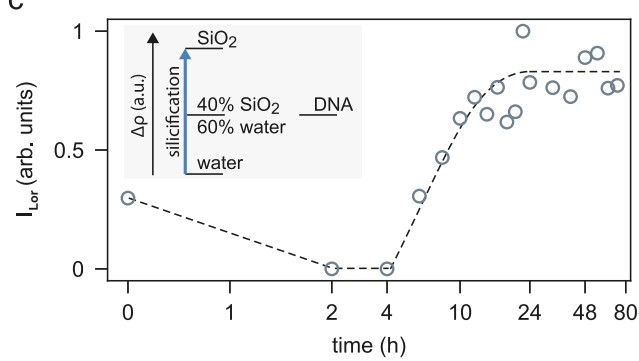

c

**Fig. 1 | In situ silicification of 24HBs while tumbling with constant speed (50 rpm) monitored by SAXS.** SAXS data is recorded for bare 24HB and during silicification (**a**). The data is shown together with the best fits of a cylinder model together with Lorentzian peaks accounting for the inner honeycomb lattice arrangement. Lorentzian peaks are highlighted by dashed lines. Data is scaled for clarity. Model-free Porod invariant Q (**b**) as a measure of the overall scattering contrast and normalized interhelical peak intensities $I_{Lor}$ (**c**) are extracted from the SAXS data shown in (**a**) as function of silica growth time. 24HB shape with honey-comb lattice structure is shown in the inset. Dashed line serves as guide to the eye.

designed DNA double helix arrangement in a honeycomb lattice, as schematically depicted in Supplementary Fig. 1a. Within this established approach, the interhelical distance was found to be $a_{bare} = 26.2 \pm 0.3$ Å. The values for $R_{bare}$ and $a_{bare}$ are in good agreement with our previously reported values for this origami type[12]. The full structure model is detailed in the Supplementary Note 3 of the supporting information.

Next, we monitored the structural changes during silicification. X-ray measurements were taken over a period of up to 80 h. Silica growth was primed by the addition of TMAPS, and subsequently initiated by the injection of TEOS (see methods for details). To determine the time required for the silicification to reach completion, we evaluated the time dependence of the Porod invariant $Q$ (Fig. 1b). Briefly speaking, if the silicification reaction yields a product that scatters more intensely than the solvent, the Porod invariant $Q$ will increase, and once the reaction stops, $Q$ will saturate. The Porod invariant $Q$ is a model-free measure of the total scattering contrast ($\Delta\rho$) of the overall sample solution, which was obtained here essentially by numerical integration of the SAXS intensities shown in Fig. 1a (see Supplementary Note 4 for details). For the bare 24HBs we obtained $Q_{bare}^{24HB}(t = 0h) = 0.3 \cdot 10^{-3}$ cm$^{-1}$ Å$^{-3}$. During silicification, $Q$ increased as a function of time. Since the electron density of amorphous silica ($\rho_{SiO2} \approx 19 \cdot 10^{-6}$ Å$^{-2}$) is larger than the electron density of water ($\rho_{H2O} = 9.4 \cdot 10^{-6}$ Å$^{-2}$), this finding is consistent with increasing silica deposition *on* or *in* the 24HBs. The Porod invariant was observed to saturate after ~24 h suggesting that the reaction had already finished at this time. This is an interesting finding since this time is much shorter than most reaction times reported previously[7,13,14,16] where reactions (employing varying reactant ratios) took up to a week. A possible explanation could be that in these reports the silicification reaction mixture was left to react undisturbed at temperatures slightly below RT, while here during the measurement gentle tumbling was applied at RT in order to avoid sedimentation. Silicification reaction kinetics are highly influenced by movement, pH and temperature, therefore tumbling at RT may have in advertedly sped up the reaction[26].

Per se, the Porod invariant is not sensitive to the distribution of the silica. Therefore, we now analyse the temporal intensity changes of the Lorentzian peak ($I_{Lor}$), which is sensitive to the inner structure of the DNA origami. Strikingly, as can be seen in Fig. 1c, this peak vanished shortly after the reaction started. However, after running the silicification reaction for more than 4 h, the Lorentzian peak recovered in intensity, surpassing the initial intensity level and even showing a second order peak at $q \approx 0.32$ Å$^{-1}$ (cf. Fig. 1a and Supplementary Fig. 5). The disappearance and recovery of a diffraction peak is a phenomenon known as contrast matching. Contrast matching occurs if the scattering length between an object and its matrix are equal[27]. The scattering length densities from water, DNA, and silica are $\rho_{H2O} = 9.4 \cdot 10^{-6}$ Å$^{-2}$, $\rho_{DNA} = 13 \cdot 10^{-6}$ Å$^{-2}$, and $\rho_{SiO2} \approx 19 \cdot 10^{-6}$ Å$^{-2}$, respectively. In turn, once ca. 40 % of the water volume fraction within the DNA origami voids are replaced by silica ($x_{SiO2} = 0.375$, compare Supplementary Note 5 in the supporting information), contrast matching occurs, i.e., the diffraction peak vanishes, as observed in Fig. 1c after 4 h. With more and more water being replaced by silica, contrast inversion, i.e., recovery of the diffracted intensity occurs as validated in Fig. 1c for later reaction times. The helix peak intensity saturated after ~24 h in accordance with the saturation of the Porod invariant $Q$, indicating completion of the silicification reaction. The occurrence of the second order peak after contrast inversion (at $q = 0.32$ Å$^{-1}$) is remarkable, since it indicates that the helical lattice is conformably coated by silica.

Previous studies on DNA origami silicification lacked information on whether silica is covering exclusively the outer surface of the DNA origami object, or penetrating the inner structure as well, embedding the individual helices[7,13,17,19]. The in situ SAXS results presented here clearly reveal that silica does form in between the double helix arrangement of the origami structure. Since the equilibrium distance of the double helix is a balance of attractive and repulsive forces, the question arises if this balance is distorted by the presence of silica. We can verify such changes by evaluating the origami cylinder radius ($R$) and the interhelical distance ($a$) of the 24HBs (cf. Fig. 2). Since TMAPS binds to the DNA backbone through electrostatic interactions, con-

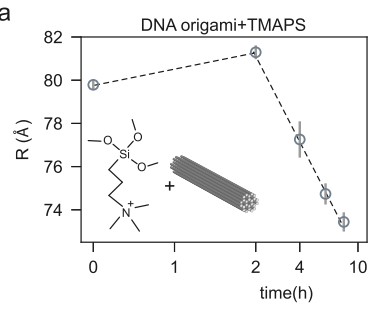

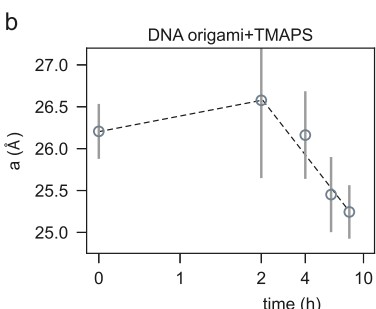

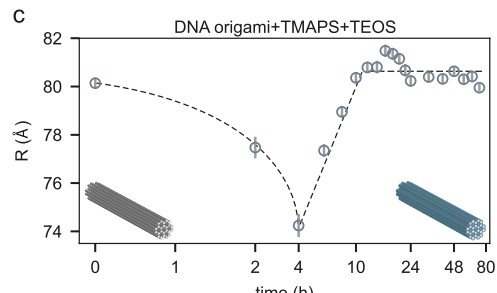

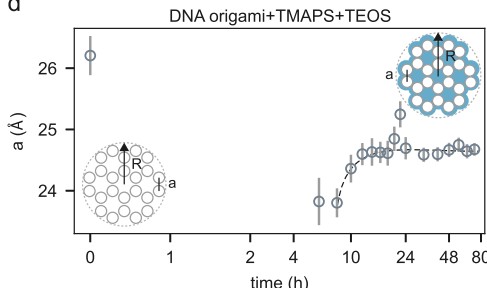

**Fig. 2 | Effect of TMAPS and TEOS on origami radius and interhelical distance.** Radii of the overall cylinder-shaped 24HBs and interhelical distance extracted from Supplementary Fig. 6 and from Fig. 1a plotted as a function of TMAPS incubation time (**a**, **b**) and of TMAPS and TEOS incubation time (**c**, **d**). Dashed lines serve as guide to the eye. Schematic of the 24HB honeycomb lattice structure are shown as insets. Error bars indicate standard deviation σ due to modelling of the x-ray data for each time point.

densation or expansion effects, as previously observed by us for change in ionic strength, or by osmotic effects, are possible[12].

So far it was unclear to which extend silicification changes the internal structure of a DNA origami. To disentangle potential effects of TMAPS and TEOS alone, bare 24HBs were incubated with TMAPS only, and studied for several hours. The corresponding SAXS data are shown in Supplementary Fig. 6. Both the 24HB cylinder radius ($R$) and inter-helical distance ($a$) show a substantial decrease in response to inter-action with TMAPS (cf. Fig. 2a, b) after an incubation time of 4 h. After 8 h, we obtained a cylinder radius of $R_{min}^{TMAPS} = 73.4 \pm 0.4$ Å and an interhelical distance of $a_{min}^{TMAPS} = 25.2 \pm 0.3$ Å. These observations indicate that the interaction of the DNA phosphate backbone with TMAPS condenses the outer radius by $6.7 \pm 0.4$ Å, and the DNA-double helix spacing by $1.0 \pm 0.3$ Å. (Longer reaction times were difficult to analyze due to an increased onset of aggregation.) Such a condensation of DNA origami objects in the early steps of silicification has never been observed before. We propose that TMAPS binding to the DNA back-bone causes electrostatic screening reducing the repulsion between neighboring helices[12,28–30], possibly in conjunction with water depletion effects. The initial lag of 4 h incubation time suggests that TMAPS accesses the phosphate groups by obstructed diffusion.

Interestingly, we observed this condensation effect even faster if TEOS was added immediately after TMAPS injection. During the first 4 h of silica growth, the cylinder radius decreased down to $R_{min} = 74.2 \pm 0.5$ Å (cf. Fig. 2c) and a strongly decreased interhelical distance $a_{min}(t = 8 h) = 23.8 \pm 0.2$ Å was observed after 8 h (cf. Fig. 2d). This accelerated condensation suggests hydrophobic effects within the origami in response to early silica formation.

A naïve comparison of the radius before ($R_{bare}$) and after silicification ($R_{SiO2}$) would suggest that there is no silica shell on the outside of the origami at all. However, since the honeycomb lattice of 24HBs remains significantly condensed even towards the end of the reaction ($a_{SiO2} = 24.7 \pm 0.05$ Å), the definition of the "outer silica shell thickness" requires some caution. We suggest that the difference between the cylinder radius at the end of the reaction ($R_{SiO2} = 80.4 \pm 0.1$ Å) to the most condensed radius ($R_{min}$) is a realistic upper limit for the silica encapsulation thickness. Here, we found $(R_{SiO2} - R_{min}) = 6.2 \pm 0.3$ Å. Thus, the outer silica shell thickness is clearly in the sub-nanometer range. Such small changes would not be detectable using conventional TEM or AFM analysis.

Silicified DNA origami show impressive thermal stability (heating up to 1200 °C)[13,14,31]. We wondered if the early condensed state of the origami with about 40 % silica infill and sub-nanometer shell already shows such enhanced temperature stability. To answer this question, we heated a DNA origami at the maximally condensed state ($R = 74.5 \pm 0.4$ Å) to 60 °C for 30 min. Bare 24HBs were already shown to fully melt between 50 and 54 °C[12]. Contrastingly, the silicified structures remained intact as confirmed by SAXS and TEM analysis (cf. Fig. 3). Surprisingly, it appears that the 40% silica frosting in the condensed origami state already provides substantial thermal stability.

All origami discussed so far were cylindrically shaped 24HBs. In order to verify our findings, we also studied cuboid, brick-shaped origami during silicification and noted a great tendency towards aggregation, which is already visible by naked eye as macroscopic clouds in solution. However, in view of the entropic forces at work during silicification, this is expected since depletion forces are best known for favoring the aggregation of colloids[32]. Since the outer coating here is subnanometer, strongly curved cylindrical origami apparently do not possess enough contact area to develop such strong aggregates. Flat surfaces of brick-like DNA structures, however, readily form aggregates. To explore this scenario on the molecular level, we investigated the silicification of a cuboid DNA origami, i.e. the 4-LB, also designed on a honeycomb lattice.

The SAXS intensity for the 4-LBs before silicification exhibits one to two distinct oscillations with dips at $q \approx 0.07$ Å$^{-1}$ and $q \approx 0.13$ Å$^{-1}$,

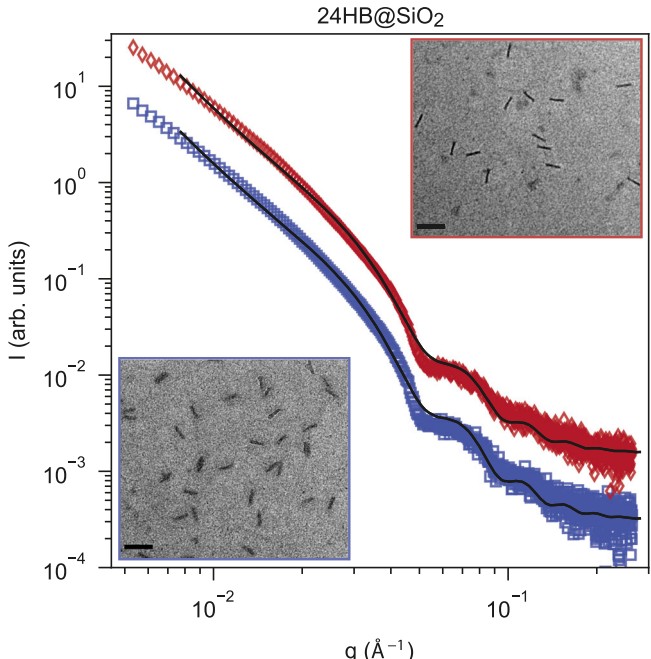

**Fig. 3 | Temperature stability of condensed silicified 24HBs verified by SAXS and TEM.** SAXS intensities of 24HBs@SiO$_2$ ($R_{SiO2} = 74.5 \pm 0.4$ Å) measured at room temperature (blue squares) and after heating the structures to 60 °C for 30 min (red diamonds) and TEM micrographs of 24HB @SiO$_2$ at room temperature (blue frame) and after heating to 60 °C for 30 min (red frame) are shown in the insets. Scale bars: 200 nm.

characteristic for the overall cuboid shape of 4-LBs, see Fig. 4a. Additionally, a pronounced Lorentzian peak arising from the honeycomb lattice design can be observed. The thickness ($A$) of the 4-LB origami is small enough to be extracted with high precision from the SAXS data of a cuboid model (cf. Fig. 4c). We obtained a thickness of $A_{bare} = 89.9 \pm 0.4$ Å. At this stage, the brick-like 4-LB origami is well dispersed, i.e., SAXS data can be modelled without the need for a structure factor.

After initiating silicification, the Porod invariant $Q$ saturates already after ~4 h, i.e., much earlier than in the case of the 24HB (cf. Fig. 4b). The overall increase of the $Q$ value after silicification is only about half compared to that of the 24HBs. During silica formation the brick thickness is condensed to a minimal thickness of $A_{min}(t = 56 h) = 80.3 \pm 1.3$ Å. However, we did not observe a reversal of the condensation effect. In agreement with this observation, the ori-gami reaches the contrast matching condition, i.e., the helix-helix peak vanished, but there is no recovery, indicating that uptake of silica is limited. Instead, we observe an upturn of SAXS intensity at small $q$-values during the 4-LBs' silica growth in Fig. 4a, which is an established fingerprint of aggregation. In some cases, this aggregation gives rise to a particle-particle stacking peak (cf. Supplementary Fig. 7). We there-fore conclude that DNA origami with flat surfaces show increased tendency towards aggregation during silicification. So somewhat paradoxically, the brick particles here form rather large aggregates without reaching similar silica uptake compared to cylindrical origami. Nevertheless, the 4-LB, similar to the 24HB showed increased thermal stability after 4 h of silicification, i.e. with an ultrathin outer silica coating, suggesting that enough silica deposition occurred to preserve the brick shape (Supplementary Fig. 8).

## Discussion

The Porod invariant $Q$ turns out to be a model-free indicator for the kinetics and yield associated with DNA origami silicification. Silicifi-cation of DNA origami is a rather slow process and the initial phase is

a

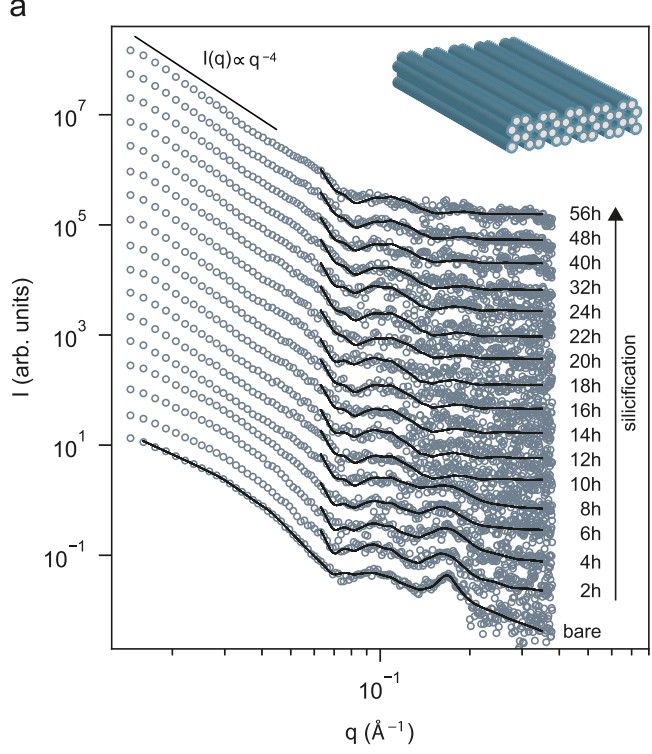

b

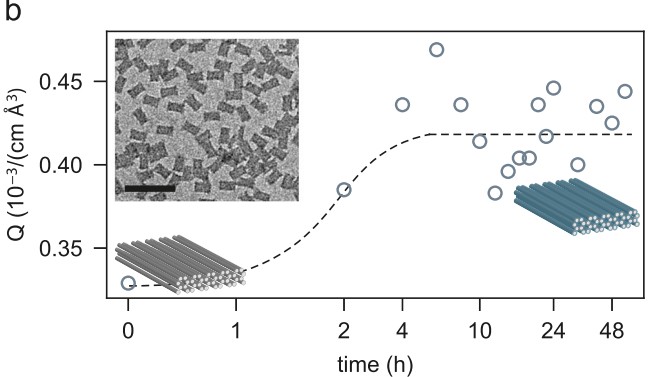

c

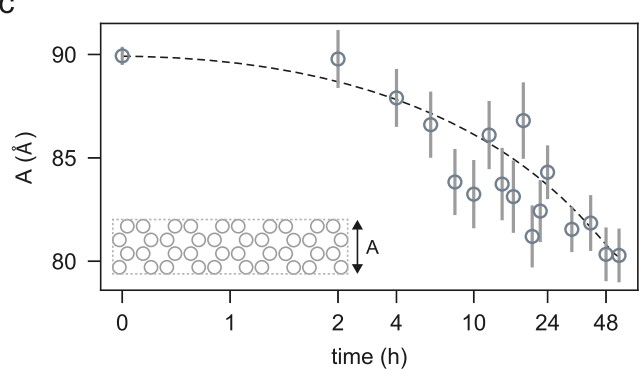

**Fig. 4 | Silicification of the 4-LB cuboids analyzed by SAXS. a** SAXS intensities of 4-LBs, before and during silicification, with best fit of a cuboid model, and Lorentzian peaks accounting for the honeycomb lattice structure. Data is scaled for clarity. **b** Silica growth time dependence of the model-free Porod invariant $Q$ extracted from **a** and a TEM micrograph of 4-LB @SiO2. Scale bars: 200 nm. **c** Heights A of the overall cuboid-shaped 4-LB as function of silica growth time. Schematic 4-LB cuboid shape with honeycomb lattice structure and front view are shown in the insets. Dashed lines serve as guide to the eye. Error bars indicate standard deviation sigma due to modelling of the x-ray data for each time point.

consumes TEOS yielding "primary silica particles", or better, short silica chains of here maybe in average 3-4 units (1 TMAPS + 2-3 TEOS = 3–4 silica units). These primary silica particles should form within minutes, i.e., much faster than the silicification reaction kinetics observed here, which takes hours. We therefore suggest that the silicification reaction of the DNA origami here is driven by phase two of the general silicification reaction; aggregation of primary silica particles and their condensation into silica networks[26,33,34]. This scenario implies the diffusion of the primary particles (silica chains) into the DNA origami and subsequent electrostatic binding of cationic TMAPS-TEOS precursors to anionic DNA. The binding of these less polar chains to the internal surfaces of DNA helices gives rise to hydrophobic effects, such as initial condensation of all of the origami structures studied. Binding to the outer surfaces favors strong aggregation of brick-shaped origami, even for ultrathin shells.

By using in situ SAXS we were able to show that a strong condensation of DNA origami nanostructures occurs during silicification. Silica deposition is not limited to the outside of the origami, but also occurs within the individual helix bundles. Interestingly, cuboidal DNA origami structures showed strong signs of aggregation during silicification and an overall decreased level of silica deposition compared to cylindrical DNA origami structures. Silica "shells" observed for both origami shapes used here are in the sub-nanometer regime, yet provide sufficient stability for shape retention at high temperatures over an extended period of time. We expect that these insights into the molecular arrangements during synthesis are key to the development of enhanced silicification protocols of DNA origami needed to fabricate e.g. sculptured dielectrics. One key requirement is to prevent aggregation of planar structures, possibly by inclusion of some bulky, water-soluble silanes, which bind only to the outer origami surface due to steric hindrance. Another aspect is that the inner part of the origami should be more readily accessible to primary silica particles to prevent their assembly outside of the origami. For this purpose, small primary particles may be explored followed by subsequent further additions of TEOS. It is well-documented that TEOS, following full or partial hydrolysis preferentially reacts with larger silica clusters rather than with itself, which, in this case, would be provided by the partially silicified DNA origami[34]. By following a careful step-by-step silicification approach, this could lead to a higher degree of control over silica shell thickness and overall structure stability.

## Methods

### Folding and purification of DNA origami structures

Both DNA origami structures used here were designed using the CaDNAno software[35] (design schematics in Supplementary Figs 1–3 and Supplementary Table 1)

24HB: The 24HB structure (design schematics in Supplementary Fig. 1a and Supplementary Fig. 2) was folded using 30 nM of DNA scaffold p8064 (tilibit nanosystems GmbH, Germany), and 100 nM of each staple oligonucleotide (Eurofins Genomics Germany GmbH and Integrated DNA Technologies, Inc., USA) in buffer containing 400 mM Tris-Acetate, 1 mM EDTA (pH = 8) and 14 mM MgCl$_2$. The mixture was heated to 65 °C and held at this temperature for 15 min, then slowly cooled down to 4 °C over a period of 15 hours[12].

characterized by a pronounced condensation upon silica incorporation, which we observe not only for origami based on honeycomb lattice arrangements, i.e. the 24HBs and 4-LBs, but also for origami structures based on a square lattice design, i.e. three-layered blocks (3-LBs) as detailed in the Supplementary Notes 8, 13, and 14. In general, silicification under similar conditions exhibits two reaction phases: Initially, TMAPS primes the silica polymerization reaction which then

The 24HBs were concentrated and purified from excess staples by two rounds of polyethylene glycol (PEG) precipitation and re-dispersion in buffer (1× TE, 3 mM MgCl₂). In brief, the origami folding solution was mixed in a 1:1 volumetric ratio with PEG precipitation buffer (15% *w/V* PEG (MW: 8,000 g/mol), 500 mM NaCl, 2× TE), adjusted to a MgCl₂ concentration of 10 mM and centrifuged at 16,000 × g for 25 min. The supernatant was removed and the DNA pellet was re-suspended in 0.5 mL of 1x TE buffer containing 11 mM MgCl₂. The PEG precipitation step was repeated after 30 min of shaking, and the purified structures were re-suspended in the final buffer (1× TE, 3 mM MgCl₂). This solution was shaken for 24 h at room temperature at 350 rpm for complete dispersion of the origami. The concentration of the purified DNA origami solution (up to 270 nM or 1.4 g/L) was verified via absorption measurements (Thermo Scientific NanoDrop 1,000 Spectrophotometer). The successful folding of structures was confirmed by TEM analysis. DNA origami solutions were stored at 4 °C until further use.

4-LB: The 4-LB (design schematics in Supplementary Fig. 1b and Supplementary Fig. 3) was folded using 10 nM of the scaffold p8064 (tilibit nanosystems GmbH, Germany), 100 nM of each staple oligo-nucleotide (Integrated DNA Technologies, Inc., USA) in buffer containing 40 mM Tris, 20 mM acetic acid, 1 mM EDTA (pH = 8) and 18 mM MgCl₂. The mixture was heated to 65 °C and held at this temperature for 15 min, then slowly cooled down to 20 °C over a period of 16 h. The 4-LB origami solution was concentrated and purified from excess staples by ultrafiltration instead of PEG purification to reduce aggregation. Briefly, the folding mixture (~2 mL) was divided over 4−5 Amicon Ultra filters (0.5 mL, 100 K, Millipore, USA) and each centrifuged at 8000× g for 8 min. The centrifugal steps were repeated 3−5 times with fresh buffer (1×TAE, 3 mM MgCl₂) added in every step. The resulting solution (~30 µL) was re-suspended in buffer and the procedure repeated. A purified origami solution of 100−120 µL in total with a concentration up to 270 nM (1.4 g/L) was obtained and stored at 4 °C until further use. The correct folding of the DNA origami was confirmed by TEM analysis

### Silica coating

110 µL of purified 24HBs (270 nM) were mixed with 0.67 µL of TMAPS (TCI, USA) (50% in methanol) and shaken at 350 rpm for 1 min in an Eppendorf tube. 2.67 µL of TEOS (Sigma Aldrich, USA) (50% in methanol) were added to the tube, followed by shaking for another 15 min. Finally, the solution was filled into a sample cell for SAXS, which tumbles slowly (50 rpm). This way, molar ratios of (1:5:12.5) of phosphate groups:TMAPS:TEOS, were achieved, respectively.

For the 4-LB structures, the TMAPS-only containing origami solution was filled into the SAXS tumbling chamber after shaking at 350 rpm for 1 min in an Eppendorf tube. Subsequently, TEOS (50% in methanol) was added 15 min later directly into the SAXS tumbling chamber and incubated directly in the sample chamber to reduce aggregation.

### TEM imaging

TEM imaging was carried out using a JEM-1230 transmission electron microscope (JEOL) operating at 80 kV. For sample preparation 5−10 µL of a solution containing (silicified) DNA origami structures were deposited on glow-discharged TEM grids (formvar/carbon-coated, 300 mesh Cu; TED Pella, Inc.) for at least 1 min, depending on sample concentration. For visualization, bare origami structures were negatively stained by briefly washing the grid with 5 µL of a 2% uranyl formate (UFO) solution followed by staining with UFO for 10−30 s. Silicified DNA origami were not stained, but washed twice with MilliQ water.

### In house SAXS experiments

Most X-ray data were recorded at an in-house Mo X-ray SAXS setup[24]. We measured at 17.4 keV X-ray energy with an X-ray beam size of 1.0 × 1.0 mm² at the sample position. Sample-to-detector distance was 1 m. Data were recorded using a Dectris Pilatus 3 R 300 K CMOS Detector (487 × 619 pixels of size (172 × 172) µm². We calibrated the sample to detector distance and the beam center position with silver behenate powder.

### Synchrotron SAXS experiments

SAXS data from 24HB@SiO₂ before and after heating of the sample solution to 60 °C for 30 min were recorded at the Austrian SAXS beamline at ELETTRA synchrotron using a beam energy of 8 keV[36], a beam size of 0.2 × 2.0 mm², and an X-ray exposure time of 12 × 10 s. The sample solution was loaded into 1.3 mm diameter quartz glass capillaries by flow-through. Sample-to-detector distance was 1.7 m. A Pilatus detector from Dectris Ltd., Switzerland with 981 × 1043 pixels of size 172 × 172 µm² served as detector. N.B.: As the SAXS chamber is an open system, heating above 60 °C would lead to significant evaporation.

## Data availability

All data supporting the key findings of this study are available within the main text and supplementary information files. The raw SAXS data generated in this study have been deposited in the Open Data LMU repository under accession code: https://doi.org/10.5282/ubm/data.315. Additional data used in this study are available from the corresponding authors upon reasonable request.

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

## Acknowledgements

We acknowledge financial support from the German Research Foundation (DFG) through SFB1032 (Nanoagents) projects A06 (A.H-J) and A07 (B.N.) number 201269156; the BMBF grant no. 05K19WMA (LUCENT), and the Bavarian State Ministry of Science, Research, and Arts grant "Solar-Technologies go Hybrid (SolTech)" provided financial support to B.N. The DFG through the Emmy Noether program (project no. 427981116) provided financial support to A.H-J. This work benefited from SasView software, originally developed by the DANSE project under NSF award DMR-0520547. We would like to thank Heinz Amenitsch for assistance in using the SAXS beamline at the synchrotron Elettra in Trieste and the CERIC-ERIC Consortium for the access to experimental facilities. Also, we would like to thank Marianne Braun for help with TEM imaging.

## Author contributions

M.O., A.H-J. and B.N. conceived the idea. M.O. fabricated samples and carried out all SAXS experiments and modeling. A.B. and L.W. designed structures, fabricated the samples and carried out analysis and microscopy. A.H-J. and B.N. supervised the study. M.O., A.H-J. and B.N. wrote the manuscript. All authors discussed and edited the manuscript and gave approval to the final version of the manuscript.

## Funding

## Competing interests

The authors declare no competing interests.
