## [Peer Review File · Nature Communications]

In situ small-angle X-ray scattering reveals strong condensation of DNA origami during silicificationREVIEWER COMMENTS

Reviewer #1 (Remarks to the Author):

In this article, Martina F. Ober and his/her coworkers reported a condensation behavior of typical honeycomb structured DNA origami during silicification process by using in situ SAXS experiments and associate data analyzing. Also, they found that silicification takes place not only on the surface, but also inside the nanopores DNA origami, based on tracing the variations of DNA double helix Lorentzian peaks at $q \approx 0.16 \text{ 1/\AA}$. Overall, this is an interesting work that provides novel insights into the DNA origami templated silicification reaction, and it is important for the understanding of the reactions on the organic-inorganic interfaces. However, the major conclusions are too preliminary based on the current data sets. The referee therefore believe that this work could not be published on Nature Communications at its current form, before the following issues have been addressed.

Please see my detailed comments below.

Major concerns

1. The authors attempted to draw a general conclusion of the condensation behavior of all DNA origami structures during silicification. But only honeycomb structured DNA origami were tested. Additional experiments on other types of DNA origami structures are necessary.
2. Rigorous SAXS data processing is particularly important for this work. However, over-fitting and fitting plots with high Chi values were observed. E.g., the fitting plots for 4-LBs sample after 10h, especially at $q \approx 0.16 \text{ 1/\AA}$, were not well agreed with the experiments. These will come to completely different conclusions for the condensation of 4-LBs. E.g., Line 272, the authors claimed that, "This aggregation may even obstruct influx of further silica particles into the origami". But aggregation of 4-LBs DNA origami should not alter the intrinsic structural characteristics of single DNA origami structure. In addition, TMPAS only control experiments were not carried out on this structure. Thus, additional experiment, analyzing and re-discussion on the 4-LBs sample group should be provided.
3. Porod invariant Q is in inverse proportion to Porod Volume. According to this, the data showed in Figure 1b suggested a reduction of Porod Volume, which, is in contradictory to the slightly increased R value showed in Figure 2c. Please check and explain this contradiction.
4. Comparing to previous studies on the sol-gel reaction of DNA origami, the authors reported a more rapid silicification reaction, and reasoned this phenomenal to the tumbling. However, their reagent ratios are different, phosphate group/TMAPS/TEOS, 1:1:15 (Previous work) and 1/5/12.5 (this work). Please provide additional explanations.

Minor comments,

1. Line 89, silica coating. The silicification reaction protocols for 24HBs and 4-LBs were not identical. Please comment on this.
2. Line 115, please provide sample-to-detector distance parameter for synchrotron SAXS experiments.
3. Line 126, the radiation damage always exists, "reduce the radiation damage" maybe a more cautious claim. Also, please provide the exposure details for each sample.
4. Line 129, "larger sample lengths", did the authors refer to "longer optical path distance"?
5. Line 230, a calculated outer silica shell with thickness of $6.2 \pm 0.3 \text{ \AA}$ was reported. Please consider the molecule size of TMAPS, length of Si-O bond, and draw a detailed molecular scheme for this layer.
6. Line 237, what will happen if we rise the temperature? And what is the critical point of this DNA-silica frosting composite?

7. Line 285, please explain why short silica chains in this experiment should have average 3-4 units and with a ratio of TEOS 2.5:TMAPS 1.
8. Line 327, Figure 1a, please highlight the Lorentzian peaks, an extra enlarged diagram at q range 0.1 to 0.2 may provide a better presentation. The second order peaks at $q \approx 0.32 \text{ 1/\AA}$ are of high interest. What is the physical meaning of these peaks? Please also explain the delayed occurring, disappearing and reoccurrence of these peaks.
9. Line 335, wrong citation of Figure S3, should be S5.
10. In Figure S5, mathematically considering, there should also have a contrast matching point for TMAPS only system, where the Lorentzian peak distinguishes. Please prove this point with additional experiments.
11. Please compare the statistical results of size information for silicified sample from TEM data with calculated R and A values.
- 12 Typos,
Numbers, e.g., 8000 should be 8,000
Please check the blanks before all units.
Please ensure the references were correctly cited. E.g., SI, Line 119, 122.

Reviewer #2 (Remarks to the Author):

The manuscript by Martina Ober and coworkers investigates silicification process of DNA origami structures by small-angle X-ray scattering. The authors observe interesting effects of DNA origami condensation and contrast inversion during the silicification process. Furthermore, the experimental results indicate that silica deposition occurs both on the outer surface and inner structure of origami constructs. This is detailed and well performed study that provides important insights on the silicification process. The manuscript is clearly written and the data is, in general, well presented. My main issue with this submission to Nature Communications is that while it is often suggested that silica coating of DNA origami structures would help to "unravel their full potential and utilization in real-life applications", in reality, the application of silica coating (which is now well-established) is limited to enhancing stability of origami-based assemblies for structural characterization. Due to this rather limited application, I am not convinced that the manuscript would be of interested to the broad readership of Nature Communications. Publication in more specialized journal seems to be more appropriate in this case.

Other comments

- TMAPS induced condensation

I am wondering why the measurements were performed only over the span of 8h (Fig 2a, b)? Since DNA origami condensation during silicification process is one of the most interesting aspects of this work, it would be important to characterize at which point the TEOS induced condensation reaches saturation. Related to this, the statement "After eight hours, we obtained a minimal cylinder radius of $73.4 \pm 0.4 \text{ \AA}$ and an interhelical distance of $25.2 \pm 210 \text{ 0.3 \AA}$ " is somewhat misleading. The cylinder radius and interhelical distance would probably still decrease with time.

- TMAPS+TEOS induced condensation

The data presented on Fig 2d. does not support the following statement "...minimal interhelical distance $23.8 \pm 0.2 \text{ \AA}$ could already be observed after 8 h". There are no measurement points between 0 and 8h, hence, it is impossible to conclude that minimum of interhelical distance is at time point of 8h.

- Aggregation.

Aggregation is one of the main problems faced during silicification of DNA origami structures in solution. The authors suggest the origami objects with flat surfaces have strong tendency towards aggregation due to entropic (depletion) forces. This is rather speculative statement; it has been observed in other studies (e.g., ref.13) that DNA origami structures have tendency for tip-to-tip stacking during silicification. As 4LB has 40 helices on the tip vs 24 helices for 24HB, the tip-to-tip stacking might be enhanced. I wonder why the authors did not provide TEM images of 4LB@SiO₂ at the early stages of aggregation, this would provide some insights on the process. Instead, the authors decided to offer rather speculative explanation for the origin of aggregation.

Minor comments

- Different folding condition and purification approaches were used for 24HB and 4LB. Why is that?

- "All origami discussed so far were cylindrically shaped 24HBs. In order to verify our findings, we also studied cuboid, brick shaped origami during silicification and noted a great tendency towards aggregation,.." was aggregation observed already at TMAPS addition step or only after addition of TEOS ?

We are very grateful for the reviewers' time and their excellent and detailed feedback. We have been able to address all points and we are confident that the manuscript is now ready for publication.

Response to Reviewer 1:

Reviewer: “In this article, Martina F. Ober and his/her co-workers reported a condensation behavior of typical honeycomb structured DNA origami during silicification process by using in situ SAXS experiments and associate data analyzing. Also, they found that silicification takes place not only on the surface, but also inside the nanopores DNA origami, based on tracing the variations of DNA double helix Lorentzian peaks at $q \approx 0.16 \text{ 1/\AA}$. Overall, this is an interesting work that provides novel insights into the DNA origami templated silicification reaction, and it is important for the understanding of the reactions on the organic-inorganic interfaces.

Response: We thank the referee for assigning novelty and importance to our report.

Reviewer: However, the major conclusions are too preliminary based on the current data sets. The referee therefore believe that this work could not be published on Nature Communications at its current form, before the following issues have been addressed.”

Response: In our point-by-point response, we include additional data (S8, S12, S13, S14) as requested by the referee and encouraged by the editor. This data reinforces our claims and hopefully lift any concerns that this work might be preliminary.

Q1> “The authors attempted to draw a general conclusion of the condensation behavior of all DNA origami structures during silicification. But only honeycomb structured DNA origami were tested. Additional experiments on other types of DNA origami structures are necessary.”

Response: We agree that DNA origami come in many shapes and it will be very important to understand if the effects that we observed here are universal. We therefore performed an additional SAXS experiment monitoring the silicification of three-layered blocks (3-LBs) with a square lattice structure (**Figure S 10**) – the design schematics, the materials and methods, the measurement, and the data analysis are now included in the supporting information, **notes S 1, S8, and S13**. Similar to the 24HB and 4-LB structures, we find a quickly increasing Porod invariant after the addition of TEOS and a substantial origami condensation during the silicification reaction for the 3-LBs (**Figure S 10 bc**).

Changes to manuscript:

SI – Page 1, 9, and 13: A new data set of cubic, brick-like structures was added to the supporting information as section S13. The shape parameters by design and the resulting fit parameter, i.e. the condensation effect, was added to the supporting information section S1 Figure S 1 and Table S 1 and section S8, Table S 4, respectively.

Main text – Line 291: We edited the following sentence in the main text “[...], which we observe not only for origami based on honeycomb lattice arrangements, i.e. the 24HBs and 4-LBs, but also for origami structures based on a square lattice design, i.e. three-layered blocks (3-LBs) as detailed in the supporting information note S8, S13, and S14.”

New Figure S 10 In-situ silicification of 3-LBs. a. SAXS data is recorded for bare 3-LBs and during silicification. SAXS intensities are shown together with the best fits of a cuboid model and Lorentzian shaped peaks accounting for the inner square lattice arrangement. Data is scaled for clarity. b. Model-free Porod invariant calculated for the data shown in (a) as function of silica growth time. c. Heights of the cuboid-shaped 3-LBs as function of silica growth time. Dashed lines serve as guide to the eye.

Q2.1> “Rigorous SAXS data processing is particularly important for this work. However, over-fitting and fitting plots with high Chi values were observed. E.g., the fitting plots for 4-LBs sample after 10h, especially at $q \approx 0.16$ $1/\text{\AA}$, were not well agreed with the experiments. These will come to completely different conclusions for the condensation of 4-LBs.”

Response: We fully agree that SAXS data need careful analysis. The analytical fit model, which we use, is quite minimalistic since it essentially reduces the origami to a simple geometric shape, scattering contrast, and regular internal structure. Therefore, the free fit is robust. The problem that the uncertainty is rather large towards longer times (silicification of 4-LBs, 10h and more) is not so much a chi square fitting problem. It is an experimental result: apparently, the 4-LB origami does not take up enough silica to induce contrast inversion reinforcing the Lorentzian peak signal. The pronounced power law behaviour observed in Figure 4a shows very clearly (and without fitting ambiguity) why this might be so. Most of the silica is apparently incorporated into large DNA origami-silica aggregates.

Changes to manuscript:

Main text – Line 274: We edited the following sentence in the main text to clarify our point: “[...] but there is no recovery, indicating that uptake of silica is limited.”

Q2.2> “E.g., Line 272, the authors claimed that, “This aggregation may even obstruct influx of further silica particles into the origami”. But aggregation of 4-LBs DNA origami should not alter the intrinsic structural characteristics of single DNA origami structure.”

Response: Driving particles in assemblies by anti-solvent effects is a widely applied concept in other field of nanoscience (Taylor *et al.*, doi.org/10.1038/s41467-021-22049-8). In this sentence, we wanted to raise awareness that a similar mechanism may be at work here as well for aggregating origami. But we agree with the reviewer that this was not studied systematically so we decided to remove this comment.

Changes to manuscript:

Main text - Line 280: We removed the sentence “This aggregation may even obstruct influx of further silica particles into the origami.”

Q2.3> “In addition, TMAPS only control experiments were not carried out on this structure. Thus, additional experiment, analysing and re-discussion on the 4-LBs sample group should be provided.”

Response: We performed an additional SAXS experiment studying the effects of TMAPS-only on bare 3-LBs as a representative for a cuboid-shaped origami structure with a square double helix arrangement (**Figure S 11**). The corresponding data is now included in the supporting information, **note S8 and S14**. The cuboid height (A) exhibits a dramatic decrease of 30% after an incubation time of 6 h (cf. **Figure S 11b**). Thus, we observe large TMAPS-induced condensation for various origami structures.

Changes to manuscript:

SI –Page 9 and 15: The new data set of TMAPS-only induced condensation of bare 3-LBs was added to the supporting information as section S14. The resulting fit parameters, i.e. the TMAPS induced condensation effect, was added to the supporting information section S8, Table S4.

Main text – Line 291: We edited the following sentence in the main text “[...], which we observe not only for origami based on honeycomb lattice arrangements, i.e. the 24HBs and 4-LBs, but also for origami structures based on a square lattice design, i.e. three-layered blocks (3-LBs) as detailed in the supporting information note S8, S13, and S14.”

New Figure S 11 SAXS intensities of bare 3-LBs and after the addition of TMAPS together with the best fits of a cuboid model and Lorentz peaks accounting for the inner lattice structure. Data is scaled for clarity. b. Heights of the cuboid 3-LBs extracted from (a) as function of TMAPS incubation time.

Q3> “Porod invariant Q is in inverse proportion to Porod Volume. According to this, the data showed in Figure 1b suggested a reduction of Porod Volume, which, is in contradictory to the slightly increased R value showed in Figure 2c. Please check and explain this contradiction.”

Response: For the purpose of the Porod analysis, the system here is a two-component mixture (Mantella *et al.*, doi.org/10.1002/ange.202004081). Component one are the silicified DNA origami and component two is the surrounding water matrix. The volume fraction of the silicified DNA is typically called ϕ_b , and we assume the referee refers to this parameter as Porod volume. In this logic the water fraction is $(1 - \phi_b)$. The scattering length are ρ_w and ρ_b for water and origami, respectively. Then the expression for Q reads

$$Q = 2\pi^2\phi_b(1 - \phi_b)(\rho_w - \rho_b)^2.$$

According to this equation, a decrease in origami volume ϕ_b could indeed reduce the Porod invariant Q. However, the SAXS data always reveal an increase of the Porod invariant and at later stages a saturation, which we interpret as end of reaction. This indicates that Q is dominated by changes of $(\rho_w - \rho_b)^2$ which increases monotonically with uptake of silica. In the supporting information, we originally only emphasized the dominant $(\rho_w - \rho_b)^2$ term. However, we see now that we should also include the volume fraction to avoid confusion.

Change to manuscript:

SI – Page 6: “For a two-phase system it is calculated via

$$Q = \int_0^\infty I(q)q^2 dq = 2\pi^2\phi_b(1 - \phi_b)\Delta\rho^2.$$

Here, the volume fraction of the silicified DNA origami is called ϕ_b and the water fraction is therefore given by $(1 - \phi_b)$. $\Delta\rho$ denotes the scattering contrast between the water and the silicified DNA origami. Thus, Q provides a measure of the volume fraction and the total scattering contrast, which is in our case the dominant contribution. Thus, monitoring Q allows to trace the silica growth.”

Q4> Comparing to previous studies on the sol-gel reaction of DNA origami, the authors reported a more rapid silicification reaction, and reasoned this phenomenal to the tumbling. However, their reagent ratios are different, phosphate group/TMAPS/TEOS, 1:1:15 (Previous work) and 1/5/12.5 (this work). Please provide additional explanations.

Response: In previous studies on the silicification reaction of DNA origami nanostructures a variety of different phosphate group:TMAPS:TEOS ratios were used (addition of TMAPS in a 1:1 to 1:8 molar ratio (phosphates:TMAPS), addition of TEOS in a 1:10 to 1:104 ratio (phosphates:TEOS)), not just 1:1:15 (Liu *et al.*, doi.org/10.1038/s41586-018-0332-7; Nguyen *et al.*, doi.org/10.1002/anie.201811323; Nguyen *et al.*, doi.org/10.1021/acs.chemmater.0c02111). We were here also inspired by the work of Wang *et al.* who showed that already a TMAPS:TEOS ratio of 1:2 can be sufficient to yield stable silicified DNA origami nanostructures (Wang *et al.*, doi.org/10.1038/s41467-021-23332-4). We have now added the additional references to the main text. In our previous work several different experimental conditions such as $[\text{MgCl}_2]$ and the movement of the sample during the reaction (static, moderate shaking) were tested and a significant difference in silicification behavior was observed (Nguyen *et al.*, doi.org/10.1002/anie.201811323). A significant silica growth on the DNA origami nanostructures and thermal stability thereof could only be observed after significantly longer reaction times compared to the silicification reactions with tumbling here in this work (thermal stability already observable after only 4 hours of silicification). In the work by Wang *et al.* ~ 10 h were required to achieve stable silicified DNA origami lattices.

Changes to manuscript:

Main text - Line 170: We have now added additional reference to the main text. “This is an interesting finding since this time is much shorter than most reaction times reported previously^{7, 13, 14, 16} where reactions (employing varying reactant ratios) took up to a week.”

Q5> “Line 89, silica coating. The silicification reaction protocols for 24HBs and 4-LBs were not identical. Please comment on this.”

Response: We thank the referee for pointing out that we were not clear enough with our description here. During the silicification of cuboid-shaped DNA origami, we observed strong aggregation (milky origami@SiO₂ solution) immediately after injection of the reaction mixture (origami + TMAPS + TEOS) into the SAXS sample chamber. We reasoned that this injection via pipette might act as an additional mixing step, which is not present in the original protocol. We therefore decided to inject the second silica precursor TEOS directly in the SAXS sample chamber as a more gentle way to mix the two. In order to clarify this point we changed the text in the manuscript accordingly.

Changes to manuscript:

Main text - Line 98: “Subsequently, TEOS (50 % in methanol) was added 15 min later directly into the SAXS tumbling chamber and incubated directly in the sample chamber to reduce aggregation.”

Q6> “Line 115 please provide sample-to-detector distance parameter for synchrotron SAXS experiments.”

Response: We now included the sample-to-detector distance (1.7 m) in the “Synchrotron SAXS experiments method” section.

Changes to manuscript:

Main text - Line 122: “Sample-to-detector distance was 1.7 m.”

Q7> “Line 126, the radiation damage always exists, “reduce the radiation damage” maybe a more cautious claim. Also, please provide the exposure details for each sample.”

Response: We clarify now that the radiation damage is drastically reduced if Mo radiation instead of Cu radiation is used. This was the case for all samples apart from the 24HB@SiO₂ heating experiment, which was measured at the synchrotron source. We now included the exposure times for the synchrotron experiments. For in house experiments, X-ray exposure times and silicification time (i.e. time points indicated in the graphs) are largely identical.

Changes to manuscript:

Main text - Line 129: “Mo X-rays induce less radiation dose compared to Cu radiation of the same intensity²⁷, allowing for long in situ SAXS experiments with drastically reduced radiation damage to the sample.”

Main text - Line 120: “[...], and an X-ray exposure time of 12 x 10 s.”

Q8> “Line 129, “larger sample lengths”, did the authors refer to “longer optical path distance”?”

Response: The reviewer is right, one could also say longer absorption length. In order to clarify, we have changed the text in the manuscript accordingly.

Changes to manuscript:

Main text – Line 132: “Furthermore, Mo radiation allows for larger absorption lengths along the beam (10 mm vs. c.a. 1.5 mm) yielding more practical geometric constrains for SAXS sample cells.”

Q9> “Line 230, a calculated outer silica shell with thickness of $6.2 \pm 0.3 \text{ \AA}$ was reported. Please consider the molecule size of TMAPS, length of Si-O bond, and draw a detailed molecular scheme for this layer.”

Response: As per the referee’s suggestion, we have now derived an estimation for the expected contour length of silica chains with 1,2,3 and 4 units taking into account all required bond lengths and bond angles. However, silica chains are unlikely fully stretched in a linear fashion, therefore the here calculated values are the absolute upper limit. If branching is taken into consideration, these values fit well with the observed outer shell thickness.

New Figure S 12 Upper image: Schematic minimal simplistic representation of a linear silica chain with four units including the bond lengths and bond angles used for a rough estimation of the length of such a fully stretched silica chain. Lower image(s): Molecular schemes of liner primary silica particles (left) and primary particles showing branching (right, different options for attached units are shown in grey).

Number of silica chain units	Estimated length of fully stretched silica chain	Estimation based on the ‘freely rotating chain’ model
1 (TMAPS only)	6.5 Å	5.1 Å
2 (TMAPS + 1 TEOS)	9.1 Å	6.0 Å
3 (TMAPS + 2 TEOS)	11.7 Å	6.8 Å
4 (TMAPS + 3 TEOS)	14.3 Å	7.5 Å

New Table S 5 Table summarizing the estimated length for fully stretched silica chains and the estimated end-to-end distance based on the freely rotating chain model for different numbers of chain units.

Changes to the manuscript:

SI - A new table S 5 with calculated values for TMAPS + one, two and three TEOS molecules including a molecular scheme was added to note S15.

Q10> “Line 237, what will happen if we rise the temperature? And what is the critical point of this DNA-silica frosting composite?”

Response: The behavior of the silicified DNA origami nanostructures at temperatures above 60 °C was not part of the current study, as we were mainly interested to see if already a minimal silica frosting could provide thermal stability as it was clearly shown in a previous study by our group (Fischer et al., [doi.org/ 10.1021/acs.nanolett.6b01335](https://doi.org/10.1021/acs.nanolett.6b01335)) that DNA origami nanostructures such as the 24HB are destroyed/melt when heated above 50-54 °C. As the SAXS chamber is open, heating above 60 °C leads to significant evaporation and does not allow for conclusive results.

Changes to the manuscript:

Main text – Lines 124 and 245: We more clearly stated why heating above 60 °C was not carried out and that bare 24HB already melt at a lower temperature (50 – 54 °C) than the 60 °C employed here. Line 124: “N.B.: As the SAXS chamber is an open system, heating above 60 °C would lead to significant evaporation”. Line 245: “Bare 24HBs were already shown to fully melt between 50 – 54°C.”

Q11> “Line 285, please explain why short silica chains in this experiment should have average 3-4 units and with a ratio of TEOS 2.5: TMAPS 1.”

Response: We thank the referee for pointing out that our reasoning might not be well enough explained. In the proposed reaction mechanism, chains out of **one** TMAPS molecule and several TEOS molecules form, whereby the number of TEOS molecules in these chains depends on the molar ratio of TMAPS:TEOS. In this case here, we suggest the formation of small primary particle chains composed of **1** TMAPS and **2 – 3** TEOS molecules. Due to the structural similarity of TMAPS and TEOS, we consider each TMAPS or TEOS molecule of a chain as one unit. This yields an average chain length of 3 – 4 units (1 TMAPS + 2-3 TEOS).

Changes to the manuscript:

Main text - Line 297: In order to clarify our point, we added the following explanation to the main text: “[...] maybe in average 3-4 units (1 TMAPS + 2-3 TEOS = 3-4- silica units).

Q12.1> “Line 327, Figure 1a, please highlight the Lorentzian peaks, an extra enlarged diagram at q range 0.1 to 0.2 may provide a better presentation.”

Response: As suggested by the referee, we highlighted the Lorentzian peaks in Figure 1a and included an enlarged view of the q-range between 0.1 and 0.4 Å⁻¹ in the supporting information (**note S 9**) to provide a better presentation of the 1st and 2nd order Lorentzian peaks (cf. **Figure S 5**).

New Figure S 5 In situ silicification of 24HBs (Figure 1a). Enlarged view of the q -range sensitive to the inner lattice design. SAXS intensities are shown together the best fits of a cylinder model together with Lorentzian peaks accounting for the inner honeycomb lattice arrangement. The first order (10) and second order (20) honeycomb lattice peaks are highlighted by dashed lines. Data is scaled for clarity.

Changes to manuscript:

Main text - Line 181: “However, after running the silicification reaction for more than 4 h, the Lorentzian peak recovered in intensity, surpassing the initial intensity level and even showing a second order peak at $q \approx 0.32 \text{ \AA}^{-1}$ (cf. Figure 1a and Figure S 5).”

SI – page 10: An enlarged diagram of the q -range between 0.1 and 0.4 \AA^{-1} was added to the supporting information as section S9, Figure S5.

Q12.2> “The second order peaks at $q \approx 0.32 \text{ 1/\AA}$ are of high interest. What is the physical meaning of these peaks? Please also explain the delayed occurring, disappearing and reoccurrence of these peaks.”

Response: In general, higher order peaks are an indication of higher crystalline order because lattice deformations generally weaken higher order peaks stronger by the well-known Debye Waller factor. Thus, one might be tempted to take these higher order peaks as an indication of a better-defined helical lattice. However, here we assume that the appearance of this higher order peaks results also from the increased contrast after inversion. The visibility of those peaks depend furthermore on the strength of other scattering contributions, which may mask higher order peaks. This way, higher order peaks may occur and disappear in the background, even if they are always present with low amplitude.

Changes to manuscript:

Main text – Line 195: In order to clarify, we add the following sentence to the main text of the manuscript: “The occurrence of the second order peak after contrast inversion (at $q=0.32 \text{ \AA}^{-1}$) is remarkable, since it indicates that the helical lattice is conformably coated by silica”

Q13> “Line 335, wrong citation of Figure S3, should be S5.”

Response: We thank the referee for pointing out this error to us. It has now been corrected in the manuscript.

Changes to manuscript:

Main text - Line 367: “Radii of the overall cylinder-shaped 24HBs and interhelical distance extracted from Figure S6 and Figure 1a as a function of TMAPS incubation time (a,b) and silica growth induced by TMAPS and TEOS (c,d).”

Q14> “In Figure S5, mathematically considering, there should also have a contrast matching point for TMAPS only system, where the Lorentzian peak distinguishes. Please prove this point with additional experiments.”

Response: The referee raises an interesting point. However, for the TMAPS-only system, there is no (or if any only very minimal) silica polymerization, i.e. only one TMAPS unit can attach to each DNA nucleotide. Therefore, the uptake is not enough to induce contrast matching.

Changes to manuscript:

SI – Page 11: In order to clarify our point, we have added an explanatory sentence to note S10, page 11: “It is noteworthy to mention here that contrast matching does not occur in the “TMAPS-only” case, as there is no formation of larger silica networks due to the absence of TEOS.”

Q15> “Please compare the statistical results of size information for silicified sample from TEM data with calculated R and A values.”

Response: Unfortunately, TEM images do not allow one to distinguish < 1 nm size differences as they were obtained from the SAXS data, which provides Å-level resolution. Changes in the interhelical distance (A) are impossible to be determined by TEM. Therefore, in the manuscript no comparisons with TEM images were made.

As per the referee’s suggestion, we have, however, attempted to obtain some statistical results for the size information for 24HBs measured from TEM images (for purpose of review only), see below. However, it must be noted that silicified structures were not stained, resulting in a lower resolution compared to stained structures, which is why size measurements are much less precise resulting in structures appearing slightly larger. Unfortunately, staining of silicified DNA origami does not result in increased resolution. Statistical information about the height of the 4LB cannot be gained from TEM data since this dimension is not visible in the TEM images.

Figure for reviewer 1: Statistical results for the diameter of the 24HB obtained from TEM images (left: bare 24HB, right: 24HB @SiO₂) and corresponding, representative enlarged TEM views of individual 24HBs (scale bars: 50 nm).

Changes to the manuscript:

Main text – Line 239: For clarification we added the following sentence to the main text: “Such small changes would not be detectable using conventional TEM or AFM analysis.”

Q16> “Typos, Numbers, e.g., 8000 should be 8,000 -> done. Please check the blanks before all units. Please ensure the references were correctly cited. E.g., SI, Line 119, 122.”

Response: We thank the referee for pointing these minor errors out to us. These have now been corrected.

Changes to the manuscript: Multiple changes throughout the whole manuscript and SI.

Response to Reviewer 2:

Reviewer: “The manuscript by Martina Ober and co-workers investigates silicification process of DNA origami structures by small-angle X-ray scattering. The authors observe interesting effects of DNA origami condensation and contrast inversion during the silicification process. Furthermore, the experimental results indicate that silica deposition occurs both on the outer surface and inner structure of origami constructs. This is detailed and well performed study that provides important insights on the silicification process. The manuscript is clearly written and the data is, in general, well presented.

Response: We’d like to thank the reviewer for finding our work interesting, detailed, well-performed, and clearly written.

My main issue with this submission to Nature Communications is that while it is often suggested that silica coating of DNA origami structures would help to “unravel their full potential and utilization in real-life applications”, in reality, the application of silica coating (which is now well-established) is limited to enhancing stability of origami-based assemblies for structural characterization. Due to this rather limited application, I am not convinced that the manuscript would be of interested to the broad readership of Nature Communications. Publication in more specialized journal seems to be more appropriate in this case.”

Response: While we agree with the reviewer that initially, the silica was only meant as a coating to stabilize structures for structural analysis, this is certainly no longer the case. While to date there have been only few reports using silicified DNA origami, these will certainly appear in the very near future. From recent conference visits it has become very obvious that there is a great interest in silicified DNA origami far beyond enabling structural characterization. A wide variety of applications ranging from lithographic masks (work by Hao Yan and team, unpublished), to templates for further material growth (unpublished work by Anton Kuzyk and team as well as very exciting already published work by the Gang group, e.g. doi.org/10.1021/acs.nanolett.0c05023 (nanostructures with highly enhanced electrical conductivity based on silicified DNA origami) or doi.org/10.1038/s41467-020-19439-9 (superconducting nanostructures based on silicified DNA origami) as already referenced in the manuscript. Our team has also recently developed a new method to retain the addressability of these silicified DNA origami, creating for the first time a fully site-specifically addressable silica nanostructure (not possible with mesoporous silica NPs for example), (*manuscript in preparation*). Therefore, we cannot agree with the referee that the work described in this manuscript would not be of interest to the broad readership of Nature Communications and we hope that the referee would agree.

Q1> “TMAPS induced condensation: I am wondering why the measurements were performed only over the span of 8h (Fig 2a, b)? Since DNA origami condensation during silicification process is one of the most interesting aspects of this work, it would be important to characterize at which point the TEOS induced condensation reaches saturation. Related to this, the statement “After eight hours, we obtained a minimal cylinder radius of $73.4 \pm 0.4 \text{ \AA}$ and an interhelical distance of $25.2 \pm 0.3 \text{ \AA}$ ” is somewhat misleading. The cylinder radius and interhelical distance would probably still decrease with time.”

Response: We fully agree with the reviewer that it would have been interesting to observe the TMAPS induced condensation for longer and therefore we did indeed follow the reaction for longer reaction times as well. However, unfortunately the data analysis became more ambiguous at later times, due to an onset of aggregation. It was not our intention to claim that the observed state was already the most minimal one, so in order to clarify this, we have omitted the word “minimal” in the text.

Changes to manuscript:

Main text – Line 214: “After eight hours, we obtained a cylinder radius of $73.4 \pm 0.4 \text{ \AA}$ and an interhelical distance of $25.2 \pm 0.3 \text{ \AA}$. (Longer reaction times were difficult to analyse due to an increased onset of aggregation.)”

Q2> “TMAPS+TEOS induced condensation: The data presented on Fig 2d. does not support the following statement “...minimal interhelical distance $23.8 \pm 0.2 \text{ \AA}$ could already be observed after 8 h”. There are no measurement points between 0 and 8h, hence, it is impossible to conclude that minimum of interhelical distance is at time point of 8h.”

Response: The referee makes a very valid point. What we wanted to say here is that after 8h, the minimum was either reached or already passed. This is a very short time in view that previous procedures run over days. In order to clarify this we have changed the wording in the main text of the manuscript.

Changes to manuscript:

Main text – Line 227 “...minimal a strongly decreased interhelical distance [...] was observed after 8 h”.

Q3.1> “Aggregation is one of the main problems faced during silicification of DNA origami structures in solution. The authors suggest the origami objects with flat surfaces have strong tendency towards aggregation due to entropic (depletion) forces. This is rather speculative statement; it has been observed in other studies (e.g., ref.13) that DNA origami structures have tendency for tip-to-tip stacking during silicification. As 4LB has 40 helices on the tip vs 24 helices for 24HB, the tip-to-tip stacking might be enhanced.

Response: Ref. 13 by Nguyen, L. et al. (doi.org/10.1002/anie.201811323) does not show any tip-to-tip stacking interactions to the best of our knowledge. Nevertheless, this was observed in ref 14 (Nguyen, M.K. et al., doi.org/10.1021/acs.chemmater.0c02111). However, we see here no order within the assembly, i.e. there is no structure factor in the X-ray data. Instead, SAXS just shows a power law behaviour, which is in agreement with the large irregular aggregation as observed by TEM as well (cf. **New Figure S 9**). Additionally, for the 4-LB we can only observe the shortest axis using SAXS (the long axis is invisible). Therefore, the observed aggregation cannot solely be due to tip-to-tip stacking.

Changes to the manuscript:

SI – Page 13: We have now included images of aggregated 4-LBs to the supporting information, note S12 and Figure S 9.

Q3.2> “I wonder why the authors did not provide TEM images of 4LB@SiO₂ at the early stages of aggregation, this would provide some insights on the process.”

Response: We tried to characterize the aggregation with TEM, but it is very difficult to obtain a representative TEM image of the early aggregation state. At later stages, aggregation is very pronounced as can be seen in the following TEM image:

New Figure S 9 Aggregated 4-LBs after silicification for > 24h with a nucleotide: TMAPS: TEOS ratio of 1:5:20. Scale bars are 200 nm.

Changes to the manuscript:

SI – Page 13: We have now included images of aggregated 4-LBs to the supporting information, note S12 and Figure S 9.

Q3.3> “Instead, the authors decided to offer rather speculative explanation for the origin of aggregation.”

Response: We believe the referee misunderstood our intentions here, which is why we would like to clarify: Since we ruled out an influence by the internal structure (cube versus hexagonal), we assessed that planar surfaces are more prone towards aggregation. Whether this is due to entropic forces or some other kind of microscopic vdW interactions between the particles was not analyzed. Indeed, any attractive short-range force will benefit from a planar interface which enables proximity. We therefore rephrased our finding more carefully.

Changes to manuscript:

Main text – Line 280: We now removed the following statement from the manuscript: “[...] possibly due to increased entropic forces.”

Q4> “Different folding condition and purification approaches were used for 24HB and 4LB. Why is that?”

Response: We carefully optimized the folding and purification conditions for each DNA origami nanostructure individually to enhance the yield of the entire procedure and avoid undesired effects such as aggregation. For the folding of each DNA origami nanostructure, a MgCl₂ concentration must be chosen that allows for the correct assembly of the nanostructure but does not lead to aggregation of the origami already during the folding procedure. While PEG precipitation proved suitable to achieve high yields for the 24HB, we noticed that ultrafiltration could help to reduce aggregation for cuboid-shaped DNA origami.

Change to manuscript:

Main text – Line 81: We have now added the following information to the main text: “[...] excess staples by ultrafiltration instead of PEG purification to reduce aggregation.”

Q5> “All origami discussed so far were cylindrically shaped 24HBs. In order to verify our findings, we also studied cuboid, brick shaped origami during silicification and noted a great tendency towards aggregation,” was aggregation observed already at TMAPS addition step or only after addition of TEOS?”

Response: The reviewer raises an important point. We indeed observed TMAPS-only induced aggregation for all investigated origami structures after 2-4 h (cf. **Figure S 6** and **Figure S 11**) and not only after the addition of TEOS. This is not surprising as TMAPS carries a net positive charge (Nguyen et al., doi.org/10.1002/anie.201811323), which is known to screen the electrostatic repulsion of the negatively charged double helices and thereby induce origami condensation (Fischer et al., doi.org/10.1021/acs.nanolett.6b01335, Roodhuizen et al., doi.org/10.1021/acsnano.9b05650). However, for the 3- and 4-LBs the aggregation induced by TMAPS+TEOS was even stronger.

Changes to manuscript:

SI – We have now added additional data for the 3-LB to the supporting information, notes S13 (Figure S 10) and S14 (Figure S 11).”

REVIEWERS' COMMENTS

Reviewer #1 (Remarks to the Author):

The revised paper is much strengthened and now appears suitable for publication in Nature Communications.

I have only one minor suggestion. In Figure S10, panel b. The scheme of DNA origami has covered the fitting plot. It would be helpful to move it away from the plot.

Reviewer #2 (Remarks to the Author):

The authors significantly improved the quality of the revised manuscript. However, I am still not convinced that this work would be of interest to the broad readership of Nature Communications. Although the study is well planned, carefully executed and nicely presented, it deals with quite technical (rather than conceptual) aspects of silicification process of DNA origami structures. As far as I can estimate, currently there are less than 10 research groups in the world that would fully appreciate such technical study of silicification process. There should be a discernible reason for a research work to deserve the visibility of publication in a Nature Portfolio journal rather than the best of the specialist journals. I just don't see such discernible reason in this case. I would be happy to support publication of this work in more specialized journal.

We are very grateful for the reviewers' time to re-assess our manuscript. We have addressed all points raised by the reviewers and the editor and we are confident that the manuscript is now ready to be accepted for publication.

Response to Reviewer 1:

Reviewer: The revised paper is much strengthened and now appears suitable for publication in Nature Communications.

Response: We thank the reviewer for acknowledging our improvements to the manuscript and for finding it now suitable for publication.

Q1> “I have only one minor suggestion. In Figure S10, panel b. The scheme of DNA origami has covered the fitting plot. It would be helpful to move it away from the plot.”

Response: We thank the reviewer for pointing this out to us – the DNA origami indeed covered the fitting plot. This has now been changed in the supporting information.

Changes to manuscript:

SI – Page 14: The silicified DNA origami schematic in Supplementary Figure 10b has been moved slightly so as to not cover the plot.

Response to Reviewer 2:

Reviewer: The authors significantly improved the quality of the revised manuscript. However, I am still not convinced that this work would be of interest to the broad readership of Nature Communications. Although the study is well planned, carefully executed and nicely presented, it deals with quite technical (rather than conceptual) aspects of silicification process of DNA origami structures. As far as I can estimate, currently there are less than 10 research groups in the world that would fully appreciate such technical study of silicification process. There should be a discernible reason for a research work to deserve the visibility of publication in a Nature Portfolio journal rather than the best of the specialist journals. I just don't see such discernible reason in this case.

I would be happy to support publication of this work in more specialized journal

Response: We thank the reviewer for acknowledging our improvements to the manuscript and finding out study well planned, carefully executed and nicely presented.

While we agree with the reviewer that our study deals with a more technical aspect, we believe that exactly this knowledge is absolutely essential for anyone wanting to work on silicified DNA origami. Right now, the DNA origami community is getting more and more determined to apply DNA silicification in various directions. Since it is still a relatively new field, having ~10 of the top research groups world-wide working on it *at this time* is already very impressive. We are confident that the deeper understanding presented here will have a positive influence on the outcome of future experiments and enable more groups to apply the method. Just at the right time.